# A Compendium of Magnetic Nanoparticle Essentials: A Comprehensive Guide for Beginners and Experts

**DOI:** 10.3390/pharmaceutics17010137

**Published:** 2025-01-20

**Authors:** Carlos O. Amorim

**Affiliations:** Physics Department and i3N, University of Aveiro, Campus de Santiago, 3810-193 Aveiro, Portugal; amorim5@ua.pt

**Keywords:** magnetic nanoparticles, nanomagnetism, magnetic materials, nanomedicine, magnetic gels, magnetic hyperthermia

## Abstract

Magnetic nanoparticles (MNPs) are advanced materials that combine the unique properties of magnetic materials and nanoscale dimensions, enabling a wide range of applications in biomedicine, environmental science, and information technology. This review provides a comprehensive yet accessible introduction to the fundamental principles, characterization techniques, and diverse applications of MNPs, with a focus on their nanoscale magnetic properties, such as superparamagnetism, single-domain behavior, and surface effects. It also delves into their classification and the critical role of parameters like magnetic anisotropy and blocking temperature. Emphasis is placed on routine characterization methods, including X-ray diffraction, electron microscopy, and magnetometry, as well as advanced concepts like magnetic hyperthermia and self-regulated heating. Designed for newcomers and experts alike, this review serves as both an educational guide and a quick-reference resource, ensuring clarity while maintaining scientific rigor.

## 1. Introduction

Magnetic nanoparticles (MNPs) are advanced materials that exhibit unique properties due to their size, typically between 1 and 100 nm, which enable a wide range of scientific and technological applications [1,2,3,4,5,6,7,8,9,10,11,12,13,14,15,16]. Their study began in the early 20th century with pioneering work by Louis Néel and Ernst Ising on the magnetic behavior of fine particles. However, it was not until the late 20th century, with advancements in synthesis and characterization techniques, that MNPs garnered significant attention for their potential applications [1,2,3,4,5,6,7,8,9,10,11,12,13,14,15,16]. Today, MNPs stand at the forefront of advanced research, finding uses in fields as diverse as biomedicine [1,5,7,8,10,11,14,15,16,17,18,19,20,21,22,23,24,25,26,27,28,29,30,31,32,33,34,35,36,37,38,39,40,41,42,43,44,45,46,47,48,49,50], environmental science [14,16,33,51,52,53,54,55,56,57,58,59,60,61,62,63,64,65], and information technology [66,67,68,69,70,71,72,73,74,75,76].

The importance of MNPs lies in their unique properties, which combine the characteristics of both magnetic materials and nanoparticles. Their minute size results in the emergence of nanoscale exclusive magnetic properties, such as superparamagnetism and superferromagnetism, while also conferring a high surface-to-volume ratio, enhancing their reactivity and enabling efficient catalysis [2,76,77,78,79,80,81,82,83,84,85] and surface functionalization with various chemical groups [3,4,6,7,8,10,13,14,33,86]. Moreover, their magnetic properties allow for manipulation by external magnetic fields, which is invaluable for applications such as targeted drug delivery [1,5,7,8,10,11,14,16,23,24,25,26,27,28,29,30,32,33,34,38,39,40,41,48,49,87], imaging techniques such as magnetic resonance imaging (MRI) [1,5,10,11,16,33,35,36,44,76], and water treatment [9,33,51,52,53,54,55,56,57,58,59,60,61,62,63,64,65].

In biomedicine, MNPs can be utilized for targeted drug delivery, enabling the precise release of therapeutics to specific tissues. Additionally, they can be incorporated into magnetic hydrogels, imparting magnetoresponsive properties that allow for the dynamic adjustment of shape, stiffness, and other mechanical and chemical characteristics. This facilitates controlled and selective drug release, minimizing side effects and enhancing therapeutic efficacy [17,18,19,20,21,22,23,24,25,26,27,28,29,30,31,37,38,39,40,41,42,43].

In environmental applications, MNPs can be employed to remove contaminants such as toxic metals, antibiotics, and pesticides from water through magnetic separation techniques [9,33,51,52,53,54,55,56,57,58,59,60,61,62,63,64,65].

Additionally, in data storage and spintronics, MNPs offer the potential to develop high-density storage media and next-generation electronic devices [66,67,68,69,70,71,72,73,74,75,76].

Despite the broad range of applications and the significant advancements in the field, MNPs remain a complex subject, especially for those new to the area. For researchers new to magnetism and nanotechnology, the plethora of concepts, equations, and methods related to MNPs can seem overwhelming. This review offers a comprehensive yet accessible introduction to MNPs, compiling essential topics to equip less experienced researchers and those transitioning from other fields with the foundational knowledge to navigate this area. Additionally, it also aims to serve as a quick-reference guide for more experienced scientists, helping them refresh key equations and concepts.

Therefore, I will focus primarily on the fundamental concepts necessary to understand the advantages and limitations of magnetic nanoparticles (MNPs), with a particular emphasis on magnetism, nanomagnetism, and magnetic materials. While topics such as synthesis methods and specific applications will be briefly addressed, they will not be covered in depth. Instead, references to comprehensive reviews on these aspects will be provided, enabling readers to explore them further based on their interests. The primary objective is to equip new researchers with the essential knowledge of magnetism and nanomagnetism, allowing them to engage with applications more effectively and better comprehend subsequent research articles and reviews.

To ensure clarity and accessibility, I will strive to simplify complex concepts without compromising scientific accuracy, providing easy-to-grasp explanations that preserve the richness of the subject, in the spirit of Albert Einstein’s famous maxim: “Make it simple, not simpler!” Additionally, this review will adopt an informal tone to make the content more engaging, reducing the entry barrier for newcomers. Finally, unless stated otherwise, I will describe quantities using the SI system.

## 2. Magnetism Fundamental Quantities

In this section, I will introduce some of the fundamental quantities and physical properties that frequently appear in the research literature on nanomagnetism. These definitions are particularly intended to guide readers who are new to this field and will also form the basis for understanding the material in the subsequent sections.

Readers may notice that, in various research papers, the magnetic field is sometimes referred to as B and, at other times, H. These terms do not merely represent different names for the same physical quantity, but rather two distinct yet related quantities.

The magnetic field B (B-field), also known as magnetic flux density or magnetic induction, is related to the magnetic field H (H-field). The latter also known as magnetic field strength or magnetizing force, through the relationship depicted by Equation (1) [88,89,90,91,92,93]:(1)$$\vec{B}=\mu_{0}\left(\vec{H}+\vec{M}\right),$$
where $\mu_{0}$ is the vacuum permeability, and $\vec{M}$ is the magnetization, a concept that I will define in more detail later. An intuitive way to understand the meaning of Equation (1) is to consider that in a material, such as a piece of iron, $\vec{H}$ represents the external magnetic field applied to the iron, which could originate from an electric current and/or a permanent magnet. Meanwhile, $\vec{M}$ denotes the magnetic response generated within the iron itself in reaction to $\vec{H}$, while $\vec{B}$ represents the resulting total magnetic field. In fact, in vacuum (M = 0), the magnetic flux density B simply relates to the magnetic field strength H by the proportional constant depicted in Equation (2) [88,89,90,91]:(2)$${\vec{B}}_{vaccum}=\mu_{0}\vec{H}.$$

Hence, in this case, B and H can be easily interchanged by applying the appropriate units: T (Tesla) for B (or G in the cgs system) and A/m (ampere per meter) for H (or Oe in the cgs system) [88,89,90,91].

To properly define magnetization, it is essential to recall that, unlike electric fields, which are generated by electric charges, there are no magnetic charges or magnetic monopoles. Instead, magnetic fields originate from moving charges/electrical currents, and, as the Lorentz force depicted in Equation (3) implies, the magnetic force results from the interaction between a magnetic field and moving charges [88,89,90,91]:(3)$$\vec{F}=q\vec{E}+q\vec{v}\times \vec{B},$$
where q is the electric charge, $\vec{E}$ the electric field, and $\vec{v}$ is the charge’s velocity.

In fact, while q is the elementary electric quantity, in magnetic theory the magnetic dipole moment $\vec{m}$ serves as the elementary magnetic quantity. This magnetic dipole moment can be generated either by a current I encircled in an area A, as in Equation (4) and illustrated in Figure 1a [88,89,90,91]:(4)$$\vec{m}=I\cdot A\cdot \hat{n},$$
where $\hat{n}$ is the unit vector perpendicular to A; or by a bar magnet, which can be visualized as two opposite fictitious magnetic monopoles separated by a distance l, as described by Equation (5) and illustrated by Figure 1b [88,89,90,91,93]:(5)$$\vec{m}=p\cdot \vec{l}.$$

In condensed matter physics, these magnetic building blocks arise primarily from the intrinsic magnetic moment of electrons known as “spin”, which can be envisioned as an infinitesimal bar magnet with a constant magnetic moment (1 Bohr magneton, $\mu_{B}$), and the orbital angular momentum associated with electrons, which, being charged, produce a magnetic moment akin to a current encircling an area. While neutrons and protons also have intrinsic magnetic moments, these contributions are generally minimal since a nucleus’ magnetic moment is two to three orders of magnitude lower than that of a single electron’s $\mu_{B}$ [88,89,90,91].

The magnetization M of a material is defined as the sum of all magnetic moments within a unit volume V, as shown in Equation (6) [88,89,90,91]:(6)$$\vec{M}=\sum_{i}\frac{{\vec{m}}_{i}}{V}.$$

M essentially quantifies the density of magnetic moments and their degree of alignment, which can vary significantly with temperature and the external applied magnetic field strength H.

The sensitivity of M to changes in an applied H field is characterized by a second-order tensor known as the magnetic susceptibility, $\chi^{m}$, defined by Equation (7) [88,89,90,92,93,94]:(7)$$\chi_{ij}^{m}=\frac{\partial M_{i}}{\partial H_{j}},$$
where i and j refer to spatial coordinates (e.g., x, y, z in Cartesian coordinates). For an isotropic material, at high temperatures and low H, the relation simplifies to its linear form, as shown in Equation (8):(8)$$\chi^{m}=\frac{M}{H}.$$

Since both M and H have units of A/m, $\chi^{m}$ (or just χ in the context of magnetic materials) is a dimensionless quantity and is often referred to as the volume magnetic susceptibility, $\chi_{v},$ due to M being the sum of m per unit of volume. Other related forms include mass magnetic susceptibility, $\chi_{v}$, and the molar mass magnetic susceptibility, $\chi_{M}$, given by the relations of Equations (9) and (10), respectively [89,95]:(9)$$\chi_{\rho }=\frac{\chi_{v}}{\rho },$$(10)$$\chi_{Mol}=\frac{Mol}{\rho }\chi_{v}=Mol\chi_{\rho },$$
where ρ is the mass density and Mol is the material’s molecular mass. Therefore, knowing the form of magnetic susceptibility in use is crucial to avoid unit inconsistencies.

Finally, by combining Equations (1) and (8), we derive Equation (11):(11)$$B=\mu_{0}\left(1+\chi \right)H,$$
which, using a similar formulation to Equation (2), results in Equation (12):(12)$$B=\mu H=\mu_{r}\mu_{0}H,$$
where $\mu =\left(1+\chi \right)$ is the material’s magnetic permeability and $\mu_{r}$ is its relative magnetic permeability.

To study magnetic materials, it is often useful to plot the magnetization M as a function of the applied field strength H. This M(H) plot reveals key properties, such as the saturation magnetization, $M_{sat}$, the remanent magnetization, $M_{r}$, also known as spontaneous magnetization, $M_{s}$, and the magnetic coercive field, $H_{c}$, as illustrated in Figure 2a. $M_{sat}$ represents the magnetization when all magnetic moments are perfectly aligned, marking the maximum possible M for the material. $M_{r}$ is the amount of magnetization that persists in the material after H is removed $\left(M_{r}=M[H=0]\right)$, while $H_{c}$ is the amount of magnetic field strength that must be applied in the opposing direction of the magnetization to demagnetize the material $\left(M[H_{c}]=0\right)$.

## 3. Types of Magnetic Materials

Magnetic materials are classified according to their intrinsic properties, the alignment of their magnetic moments in response to an external magnetic field, and their magnetization behavior as a function of temperature. In this section, I will provide a concise overview of the most prominent and commonly encountered types of magnetic materials, namely diamagnetic (DM), paramagnetic (PM), ferromagnetic (FM), antiferromagnetic (AFM), and ferrimagnetic (FiM).

### 3.1. Diamagnetic Materials

In diamagnetic materials (e.g., Cu, Au, Ag, Bi, graphite, diamond, and polymers such as PTFE, PVC, and PMMA), the atoms or ions lack permanent magnetic moments, as they do not have unpaired electrons. As a result, in the absence of an external magnetic field, their magnetization is zero ($M_{r}=0$ and $H_{c}=0$). However, when an external magnetic field is applied, it induces temporary magnetic dipole moments, which produce a weak induced magnetic field that opposes the applied field. This results in a very weak negative magnetization, as illustrated in Figure 2b [88,89,91,92,96].

The magnetic susceptibility of DM materials ($\chi_{DM}$) is generally small, negative, and constant, within the range of $-1\ll \chi_{DM}<0$, as shown in Figure 3b [95]. Diamagnetism arises primarily from induced currents in closed atomic orbitals, making it, in a first approximation, temperature independent, as shown in Figure 4. It is important to note that, technically, all materials exhibit some degree of diamagnetism. However, when atoms or ions in the material possess non-zero magnetic moments, other magnetic effects usually dominate, overshadowing the diamagnetic contribution [88,89,91,92,96].

### 3.2. Paramagnetic Materials

Paramagnetic materials (e.g., Al, W, Pd, O_2_, FeCl_3_ and Gd_2_(SO_4_)_3_) contain atoms or ions with permanent magnetic dipoles moments, typically due to unpaired electrons. However, in the absence of an external magnetic field, these dipoles are randomly oriented, resulting in nearly zero net magnetization, as illustrated in Figure 2c [88,89,91,92,94,96].

When an external magnetic field is applied, these dipoles tend to align in the direction of the field, resulting in a positive magnetization. The magnitude of this magnetization depends on two competing factors: Zeeman’s magnetic energy, which promotes alignment of the dipoles along the applied magnetic field, and thermal energy, which fosters the randomization of the dipole orientations. This interplay can be described by Equation (13) [88,89,91]:(13)$$M=M_{sat}B_{J}\left(z\right)=n\cdot g_{J}\cdot J\cdot \mu_{B}\cdot B_{J}(z),$$
where n is the number of magnetic dipole moments per unit of volume, $\mu_{B}$ is the Bohr magneton, and J=L+S represents the total angular momentum quantum number, combining the orbital angular momentum (L) and the spin angular momentum (S) contributions for each atom or ion’s magnetic moment. The gyromagnetic ratio, or Landé g-factor, $g_{J}$, is given by Equation (14) [88,89,91]:(14)$$g_{J}=g_{L}\frac{J\left(J+1\right)-S\left(S+1\right)+L(L+1)}{2J(J+1)}+g_{S}\frac{J\left(J+1\right)-S\left(S+1\right)+L(L+1)}{2J(J+1)},$$
where $g_{L}=1$ and $g_{S}\approx 2$ are the orbital and spin g-factors, respectively. Lastly $B_{J}$ is the Brillouin function defined by Equation (15) [88,89,91]:(15)$$B_{J}\left(z\right)=\frac{2J+1}{2J}coth\left(\frac{2J+1}{2J}z\right)-\frac{1}{2J}coth\left(\frac{1}{2J}z\right),$$
where z is the ratio between Zeeman and thermal energies, as defined by Equation (16):(16)$$z=\frac{E_{Zee}}{E_{T}}=\frac{g_{J}\cdot J\cdot \mu_{B}\cdot \mu_{0}H}{k_{B}T}$$
where H is the magnetic field strength, $k_{B}$ is the Boltzmann constant and T is the temperature.

Figure 3 illustrates the M(H) dependence of a PM material with J=12 (e.g.,: a Cu^2+^ based salt) and J=52 (e.g., a Fe^3+^ based salt) at 2 and 300 K for a constant density of magnetic moments, n. Despite being possible to saturate a PM material using high enough magnetic fields or at low enough temperatures, in typical laboratorial conditions, at room temperature, the thermal energy typically dominates, keeping the system in a nearly linear, low-z regime. In this regime, the magnetic susceptibility $\chi_{PM}$ can be approximated to a small and almost H-constant magnetic susceptibility, $0<\chi_{PM}\ll 1$, as defined by Equation (8) [88,89,91,94,95,96].

On the other hand, even at this small z regime, $\chi_{PM}$ is highly temperature dependent, following the Curie Law, as defined by Equation (17) and illustrated in Figure 4 [88,89,91,94]:(17)$$\chi_{PM}\left(T\right)=\frac{M\left(T\right)}{H}=\frac{C}{T},$$
where C is the Curie constant, which can be defined by Equation (18):(18)$$C=\frac{n\cdot \mu_{0}\cdot {g_{J}}^{2}\cdot {\mu_{B}}^{2}\cdot J(J+1)}{3k_{B}}.$$

### 3.3. Ferromagnetic Materials

Ferromagnetic materials (e.g.,: Fe, Ni and Co) are characterized by a long-range magnetic order in which magnetic dipole moments tend to align in the same direction, even in the absence of an external magnetic field (Figure 2d). The simplest microscopic description of this alignment tendency is provided by the Heisenberg model, which states that neighboring magnetic moments interact through an exchange interaction, as expressed in Equation (19) [88,89,90,92,97,98]:(19)$$H_{ex}\equiv E_{ex}=-\sum_{i\neq j}^{N}J_{ij}{\vec{S}}_{i}\cdot {\vec{S}}_{j},$$
where $H_{ex}$ is the Heisenberg Hamiltonian, essentially representing the energy $E_{ex}$ associated with the exchange interaction. ${\vec{S}}_{i}$ is the spin vector at site i, and $J_{ij}$ is the exchange constant (or exchange integral), which quantifies the strength of the interaction between the spins at sites i and j.

Equation (19) clearly shows that the dot product between spins minimizes $E_{ex}$ when the magnetic moments at sites i and j are either parallel or antiparallel, depending on the sign of $J_{ij}$. In FM materials, $J_{ij}>0,$ which creates a preference for aligned magnetic moments, leading to a spontaneous net magnetization and giving FM materials their distinctive, strong magnetic properties, as illustrated in Figure 3a [88,89,90,92,96,97,98].

However, as in PM materials, other energy terms compete with $E_{ex}$, particularly thermal energy at non-zero temperatures. A simplified way to capture the influence of $E_{Zee}$, $E_{T},$ and $E_{ex}$ is to adapt the PM framework by introducing an effective magnetic field generated by the spontaneous magnetization of the material. Pierre Weiss was the first to propose replacing the applied magnetic field with an effective field $H_{\mathrm{eff}}$, as described in Equation (20) [88,89,90]:(20)$$H_{\mathrm{eff}}=H_{\mathrm{app}}+H_{mol}=H_{\mathrm{app}}+\lambda M,$$
where $H_{\mathrm{app}}$ is the external applied field, and $H_{mol}$ is the Weiss molecular field, which is proportional to the material M by a constant λ. Substituting this $H_{\mathrm{eff}}$ into the expression for magnetization presented in Equation (13) yields the mean-field description of the FM behavior presented in Equation (21):(21)$$M=M_{sat}B_{J}\left(\frac{g_{J}\cdot J\cdot \mu_{B}\cdot \mu_{0}}{k_{B}T}\left(H_{app}+\lambda M\right)\right).$$

Solving this transcendental equation allows us to properly represent the field and temperature dependencies of FM materials. As depicted in Figure 4, a FM material exhibits weak temperature dependence at low temperatures, with magnetization remaining nearly constant until a critical temperature called Curie temperature ($T_{C}$) is reached. Beyond $T_{C}$, magnetization rapidly decreases as thermal energy overcomes the exchange interaction energy, causing the material to behave like a PM material. In the low-z regime, it is possible to describe the behavior of a FM material above $T_{C}$, hence in its PM state, using the Curie–Weiss law, as described in Equation (22) [88,89,90,92,94,96]:(22)$${\left[\chi_{FM}\left(T\right)\right]}_{T>T_{C}}={\left(\frac{M\left(T\right)}{H}\right)}_{T>T_{C}}=\frac{C}{T-\Theta_{p}}=\frac{C}{T-T_{C}}=\frac{C}{T-\lambda C}.$$

For $T<T_{C}$, Equation (23) is a simpler alternative mean-field approach to describe M as a function of temperature [88,89]:(23)$${\left(M\left(T\right)\right)}_{T<T_{c}}=M_{sat}{\left(1-\frac{T}{T_{C}}\right)}^{\beta },$$
where beta is a critical exponent.

Lev Landau demonstrated that close to $T_{C}$, β=12 effectively captures the temperature dependence of M. Additionally, Felix Bloch showed that at sufficiently low temperatures, spin waves (or magnons) significantly influence magnetization in ferromagnets, resulting in the Bloch T32 law depicted in Equation (24) [89,99,100,101]:(24)MT=Msat1−TTC32.

An empirical interpolation between the two regimes is provided by Equation (25) [101]:(25)$$M\left(T\right)=M_{sat}{\left[{1-\left(\frac{T}{T_{C}}\right)}^{\alpha }\right]}^{\beta },$$
which was proposed by several authors. Within this spirit, a popular fitting model for experimental M(T) curves of FM materials is given by Equation (26) (Figure 4) [101,102,103]:(26)MT=Msat1−sTTC32−(1−s)TTC52β,
where s is a dimensionless parameter that shapes the curve, constrained within 0<s<52, and the critical exponent β is often considered as β=0.369 to comply with the critical exponent predicted by the 3D Heisenberg model.

### 3.4. Antiferromagnetic Materials

In antiferromagnetic materials (e.g.,: MnO, NiO, and hematite (α-Fe_2_O_3_)), neighboring magnetic moments tend to align in opposite directions, resulting in no net magnetization in the absence of an external magnetic field (Figure 2e). This alternating spin arrangement can be understood through the Heisenberg model described in Equation (19), but here, the antiferromagnetic material has $J_{ij}<0,$ which favors the antiparallel alignment of its magnetic moments [4,88,89,90,91,96].

Antiferromagnetic materials can also be described through a molecular field model involving two opposing magnetic lattices, $M_{A}$ and $M_{B}$, each with their own effective field, $H_{\mathrm{eff}}^{A}$ and $H_{\mathrm{eff}}^{B}$, defined by Equations (27) and (28) [88,89,90]:(27)$$H_{\mathrm{eff}}^{A}=H_{\mathrm{app}}+\lambda_{AA}M_{A}+\lambda_{AB}M_{B},$$(28)$$H_{\mathrm{eff}}^{B}=H_{\mathrm{app}}+\lambda_{BB}M_{B}+\lambda_{AB}M_{A},$$
where $\lambda_{AA}$ and $\lambda_{BB}$ are the intra-sublattice molecular field interactions, and $\lambda_{AB}$ is the inter-sublattice molecular field coupling. The total magnetization becomes [88,89,90]:(29)$$M={M_{sat}}_{A}B_{J_{A}}\left(\frac{g_{J_{A}}\cdot J_{A}\cdot \mu_{B}\cdot \mu_{0}}{k_{B}T}\left(H_{app}+\lambda_{AA}M_{A}+\lambda_{AB}M_{B}\right)\right)\;\;\;\;\;\;\;\;\;\;\;\;\;\;\;-{M_{sat}}_{B}B_{J_{B}}\left(\frac{g_{J_{B}}\cdot J_{B}\cdot \mu_{B}\cdot \mu_{0}}{k_{B}T}\left(H_{app}+\lambda_{BB}M_{B}+\lambda_{AB}M_{A}\right)\right).$$

For pure AFM materials, where the two magnetic sublattices are identical, their magnetizations cancel out at H=0, leading to $H_{c}=0$ and $M_{r}=0$.

AFM materials exhibit a linear response to an applied magnetic field (Figure 3b) until a critical field strength, known as the spin-flop transition, where spins reorient slightly, resulting in a weak net magnetization [88,89,90,91,95].

The Curie–Weiss law can also describe the PM state of an antiferromagnetic material, using a Néel temperature, $T_{N}$, rather than a $T_{C}$. Above $T_{N},$ the critical temperature is negative ($\Theta_{p}<0$), as shown in Equation (30) (Figure 4) [88,89,90,94,96]:(30)$${\left(\chi_{AFM}\left(T\right)\right)}_{T>T_{N}}={\left(\frac{M\left(T\right)}{H}\right)}_{T>N}=\frac{C}{T-\Theta_{p}}=\frac{C}{T+T_{N}}=\frac{C}{T-C(\lambda_{AA}+\lambda_{AB})}.$$

### 3.5. Ferrimagnetic Materials

Ferrimagnetic materials (e.g.,: Fe_3_O_4_ (magnetite), γ-Fe_2_O_3_ (maghemite), Y_3_Fe_5_O_12_ (YIG) and transition metal ferrites (MFe_2_O_4_) such as CoFe_2_O_4_, NiFe_2_O_4_ and MnFe_2_O_4_) are a subclass of AFM materials, where the opposing magnetic moments differ in magnitude, resulting in a net magnetization (Figure 2e) [88,89,90,91].

Although ferrimagnets are technically a type of antiferromagnet, their response to an applied magnetic field resembles ferromagnets more closely, but with slightly weaker magnetization due to partial cancelation of opposing moments (Figure 3a) [88,89,90,91].

Similarly, FiM materials temperature dependence is also similar to the one observed in ferromagnets, also exhibiting a Curie temperature, above which they become paramagnetic (Figure 4). Some ferrimagnets show a compensation temperature where net magnetization vanishes as opposing moments cancel out due to differing temperature dependencies in the sublattices [88,89,90,91].

These classifications (diamagnetic, paramagnetic, ferromagnetic, antiferromagnetic, and ferrimagnetic) reveal unique magnetic behaviors dependent on internal interactions and external conditions like temperature and applied fields. Understanding these distinctions is crucial for applications across technology, biomedicine, and materials science.

## 4. Nanomagnetism

### 4.1. Magnetic Domains

In the previous sections, I established the fundamental principles of magnetic properties and reviewed the primary magnetic materials and their dependencies on temperature and magnetic field strength.

In the context of nanomedicine and most nanomagnetism applications, the most relevant materials are FM and FiM due to their strong magnetic field dependence and weak temperature dependence below and far from their Curie temperatures. These properties make FM and FiM materials the preferred choice for applications involving strong magnetic interactions.

While the description of FM and FiM properties of previous sections is broadly accurate, it does not capture all the complexity inherent to real materials. In some cases, additional concepts are needed to fully understand the magnetic properties of an actual material. One example is the formation of magnetic domains in FM and FiM materials. For simplicity, I will focus the following discussion about magnetic domains on FM materials, though the concept applies similarly to FiM materials.

Magnetic domains form because, in a uniformly magnetized material, the material itself generates a magnetic field known as a stray field or demagnetizing field, as illustrated in Figure 5a. This field will increase the magnetostatic energy of the whole system by an energy per unit volume, $\varepsilon_{mag}$ [76,88,89,90]:(31)$$\varepsilon_{mag}=\frac{E_{mag}}{V}=\frac{1}{2}\frac{B^{2}}{\mu_{0}}.$$

To minimize magnetostatic energy, the system reduces the stray field by forming multiple opposing magnetic domains, as illustrated in Figure 5b,c [1,4,32,76,87,89,90,93,97,98,104].

In FM materials, neighboring magnetic moments naturally align in parallel due to exchange interactions (as described by Equation (19) and illustrated in Figure 2d). However, with the formation of magnetic domains, the edge moments of adjacent oppositely aligned domains would be antiparallel (Figure 5b), which, based on the Heisenberg model, incurs an energy cost. Nature minimizes this energy cost by forming domain walls, which allow a gradual reorientation between neighboring magnetic moments, as depicted in Figure 6. There are two types of domain walls in FM and FiM materials: the Néel wall (Figure 6a) and the more common Bloch wall (Figure 6b) [88,89,90,93,97,98].

However, the formation of magnetic domains also contracts additional energy costs, primarily due to magnetic anisotropy, the material’s tendency to align its magnetization along a preferred direction, known as the easy axis. As a first approximation, we can describe the magnetic anisotropy energy, $E_{MAE}$, using Equation (32) [1,4,32,76,87,89,90,93,97,98,104]:(32)$$\frac{E_{MAE}}{V}=K_{a}{sin}^{2}\theta ,$$
where $K_{a}$ is magnetic anisotropy constant, and θ is the angle between the direction of M and the easy axis. The total magnetic anisotropy arises from multiple contributions, namely from shape anisotropy and magnetocrystalline anisotropy ($K_{a}=K_{u}$), with the latter having the greatest impact on the energy cost of magnetic domain walls formation.

In fact, the formation of a domain wall results from the balance between exchange and magnetic anisotropy energies. For a Bloch wall, the energy cost per unit of area, $\sigma_{wall}$, is approximately [1,4,32,76,87,89,90,93,97,98,104]:(33)$$\sigma_{wall}=\frac{E_{wall}}{A}\approx \pi \sqrt{\frac{K_{u}\cdot 2J\cdot S^{2}}{a}},$$
and the domain wall width $\delta_{wall}$ is given by(34)$$\delta_{wall}\approx \pi \sqrt{\frac{2J\cdot S^{2}}{K_{u}\cdot a}},$$
where $K_{u}$ is the magnetocrystalline anisotropy constant, A is the wall area, and J, S, and a are the material’s exchange integral, spin vector magnitude, and lattice parameter, respectively. Thus, the width of a Bloch wall increases with higher values of J and decreases with higher values of $K_{u}$. Equations (33) and (34) can also be expressed using the exchange stiffness, $A_{ex}$, defined by Equation (35) [1,4,32,76,87,89,90,93,97,98,104]:(35)$$A_{ex}=z\frac{J\cdot S^{2}}{a},$$
where z is the number of nearest neigbours. For a simple cubic crystal, the domain wall energy areal energy density and width can be rewritten as [1,4,32,76,87,89,90,93,97,98,104](36)$$\sigma_{wall}=\frac{E_{wall}}{A}=\pi \sqrt{A_{ex}\cdot K_{u}},$$(37)$$\delta_{wal}=\pi \sqrt{\frac{A_{ex}}{K_{u}}}.$$

The creation of these magnetic domains and domain walls can also be influenced by other micromagnetic factors, which ultimately affect the $H_{c}$ and $M_{r}$ of the magnetic material under study [1,4,32,76,87,89,90,93,97,98,104].

### 4.2. Single Domain MNPs

Building upon the concept of magnetic domains discussed earlier, let us consider FM or FiM MNPs with dimensions comparable to the domain wall width. For such small particles, it becomes energetically unfavorable to form multiple magnetic domains, since the domain walls, governed by Equations (36) and (37), cannot fit within the NP. Consequently, a critical diameter exists below which the particle can sustain only a single magnetic domain, as depicted in Figure 7 [15,37,49,86,87]. This critical diameter can be estimated using Equation (38) [1,4,32,76,87,89,90,93,97,98,104]:(38)$$d_{crit}\approx \frac{18\sqrt{A_{ex}K_{u}}}{\mu_{0}M_{sat}^{2}}.$$

### 4.3. Superparamagnetism

At ${T\ll T}_{C}$, for MNPs with ${d\leq d}_{crit}$, the magnetic moment orientation can fluctuate over time due to thermal energy. Each nanoparticle’s magnetic moment has two equivalent energy minima, corresponding to orientations along or against an arbitrary reference axis. These minima are separated by an energy barrier $K_{\mathrm{eff}}$, where $K_{\mathrm{eff}}$ is an effective magnetic anisotropy constant, with $K_{\mathrm{eff}}\approx K_{u}$ for spherical particles. Thermal fluctuations cause Néel relaxations (Figure 8), with a characteristic relaxation time $\tau_{N}$ given by Equation (39) [1,4,32,36,45,46,47,76,89,90,92,97,98,100,105]:(39)$$\tau_{N}=\tau_{0}e^{\left(\frac{K_{\mathrm{eff}}V}{k_{B}T}\right)},$$
where $\tau_{0}$ is the attempt time, given explicitly by Equation (40) [1,106]:(40)$$\frac{1}{\tau_{0}}=\frac{4}{3}\frac{q_{e}K_{u}}{m_{e}M_{sat}}\left|3G\lambda +\mu_{0}DM_{sat}^{2}\right|\sqrt{\frac{2V}{\pi Gk_{B}T}},$$

with $q_{e}$ and $m_{e}$ being the charge and mass of the electron, respectively, G the shear modulus, λ the longitudinal saturation magnetostriction, and D a numerical coefficient ranging from $\frac{4}{5}\pi$ for spherical particles to π for cylindrical particles, with typical $\tau_{0}$ ranging from $\approx 10^{-12}$ to $\approx 10^{-9}\;$s. Moreover, although $\tau_{0}$ depends on temperature, its variation is minimal, hence it is typically considered temperature-independent.

If $\tau_{N}$ is comparable or shorter than the measurement/observation time, $\tau_{m}$, the MNP’s magnetic moment fluctuates, resulting in zero spontaneous magnetization. In this regime, the magnetization of an assemble of such MNPs resembles that of a paramagnetic PM material, where each MNP behaves as a giant atom, with $J_{MNP}=N\cdot J_{atom}$, N being the total number of atoms within the MNP, and $J_{tom}$ is the total angular momentum quantum number. This behavior, observed in non-interacting MNPs with $\tau_{N}≲\tau_{m}$, is known as superparamagnetism (SPM) [1,4,32,36,45,46,47,76,89,90,92,97,98,100,105].

The magnetization dependence on the applied magnetic field and temperature for SPM materials can be described using Brillouin formalism by substituting J. with $J_{MNP}$ in Equation (13). Figure 9 illustrates the magnetization of SPM MNPs with two different diameters, compared to that of a simple PM material. As shown in Figure 9, increasing the diameter of the MNPs results in a higher number of magnetic moments and increases $J_{MNP}$, making the MNPs more easily magnetizable due to the larger $J_{MNP}$ [4,36,49,88,89,90,91,93,97,98].

For J→∞, the Brillouin function reduces to the Langevin function, $L\left(z\right)$, as expressed in Equation (41) [1,4,32,36,45,46,47,76,89,90,92,97,98,100,105]:(41)$$M(H,T)=M_{sat}L\left(z_{SPM}\right)=M_{sat}\left(\coth\left(z_{SPM}\right)-\frac{1}{z_{SPM}}\right),$$
where $z_{SPM}$ is given by Equation (42):(42)$$z_{SPM}=\frac{m_{MNP}\cdot \mu_{0}H}{k_{B}T},$$
and $m_{MNP}$ is the total magnetic moment of the whole MNP.

In contrast, if $\tau_{N}\gg \tau_{m}$, the magnetic moment orientation of the MNPs appears frozen over the measurement/observation timescale. Consequently, these MNPs are in a blocked state, exhibiting spontaneous magnetization and strong responsiveness to an applied magnetic field strength, like in the FM behavior (Figure 9) [1,4,32,36,45,46,47,76,89,90,92,97,98,100,105].

As demonstrated in Equation (39), the value of $\tau_{N}$ will greatly depend on temperature. Thus, a blocking temperature $T_{B}$ can be defined as the temperature below which the MNPs remain in a blocked state, as expressed in Equation (43) [1,4,32,36,45,46,47,76,89,90,92,97,98,100,105]:(43)$$T_{B}=\frac{K_{\mathrm{eff}}V}{k_{B}ln\left(\frac{\tau_{m}}{\tau_{0}}\right)}\approx \frac{K_{\mathrm{eff}}V}{25k_{B}},$$
where the approximation on the right-hand side assumes $ln\left(\frac{\tau_{m}}{\tau_{0}}\right)\approx 25$, a common criterion to ensure that $\tau_{N}\gg \tau_{m}$. This corresponds to typical measurement/observation times of 10–100 s, standard for magnetometry measurements, and $\tau_{0}$ values in the range of $10^{-10}–10^{-9}$ s, which are typical for most materials [1,4,32,36,45,46,47,76,89,90,92,97,98,100,105,106].

Similarly, for a given temperature, the minimum size for a SPM NP can be estimated using Equation (44):(44)$$d_{SPM}=\sqrt[{3}]{\frac{6k_{B}T\cdot ln\left(\frac{\tau_{m}}{\tau_{0}}\right)}{\pi \cdot K_{\mathrm{eff}}}}\approx \sqrt[{3}]{\frac{150k_{B}T}{\pi \cdot K_{\mathrm{eff}}}}.$$

SPM is a fascinating type of magnetism with significant applications in nanomedicine and environmental remediation [1,2,3,4,5,6,7,8,9,34,57,58,59,60,61,62,63,64,65]. SPM NPs can be easily magnetized using relatively small applied magnetic fields, yet they exhibit no $H_{C}$ and zero $M_{r}$ once the field is removed, as illustrated in Figure 9. This lack of residual magnetism prevents the agglomeration of SPM NPs due to magnetic attraction, a critical feature in biomedical applications where unwanted clustering could block blood vessels and lead to strokes or embolisms.

Careful control of MNP size is crucial not only to avoid such risks but also to optimize their magnetic properties. Larger MNPs possess higher $J_{MNP}$, making them easier to magnetize, as shown in Figure 9. For applications requiring SPM behavior, MNPs should be as large as possible (within application constraints) without entering the blocked state [88,89,90,93,97,98].

On the contrary, for magnetic storage applications, where each MNP functions as an individual bit, maximizing storage density requires blocked MNPs that are as small as possible. In such cases, SPM is undesirable because it would lead to the spontaneous loss of stored data [68,69,70,71,72,73,74,75].

### 4.4. Superferromagnetism

In this manuscript, all considerations regarding MNPs assume that each MNP does not interact with others, thereby neglecting dipolar interactions. This approximation holds true for dispersed or diluted MNPs. However, when MNPs are closely packed, their magnetic moments can interact via dipole–dipole interactions. This interaction can lead to a collective alignment of magnetic moments, even if each particle remains individually in an SPM state.

The resulting alignment produces a net magnetization across the particle assembly, resembling the FM behavior. This phenomenon was coined as superferromagnetism. However, unlike true SFM, superferromagnetism arises not from electronic exchange coupling but from long-range dipolar interactions between distinct particles.

### 4.5. Most Popular MNPs

While, in principle, any ferromagnetic (FM) or ferrimagnetic (FiM) material can be used to produce MNPs, in practice, a limited selection of materials is commonly chosen due to their favorable properties and practical considerations. These materials fall into two main categories: oxide-based MNPs and metallic MNPs.

#### 4.5.1. Metallic MNPs

The most common metallic MNPs are elemental NPs such as Fe, Ni, and Co, valued for their high $M_{sat}$ and $T_{C}$, and strong ferromagnetic properties. However, metallic MNPs are prone to oxidation, and in the case of Ni and Co, toxicity concerns require proper passivation layers to address both stability and safety issues [7,31,76,81,83,87,92,96,107,108].

FePt and CoPt alloy MNPs are also used in specific applications like high-density data storage, where exceptionally high magnetic anisotropies are necessary. Despite their potential, these alloy MNPs are less commonly used due to their higher cost and complex synthesis [66,67,68,69,72].

#### 4.5.2. Oxide Based MNPs

Oxide-based MNPs, in contrast, are chemically mote stable and easier to produce, often through cost-effective and scalable methods such as co-precipitation or hydrothermal synthesis, for example [1,5,6,8,10,11,12,13,16,32,34,36,48,76,86,87,92]. Among these, ferrites are particularly popular, with iron oxides like magnetite and maghemite being the most widely used. These materials exhibit high $M_{sat}$, elevated $T_{C}$, and strong FiM properties, while also benefiting from relatively low toxicity and the abundance of their constituent elements [14,15,16,36,48,61,92,96,109,110].

### 4.6. Surface Effects

Surface effects are pivotal in determining the properties of MNPs due to their inherently high surface-to-volume ratio. This effect becomes increasingly significant as the size of the MNP decreases, with surface atoms/ions comprising a substantial fraction of the total particle composition. This dominance of surface atoms introduces distinct magnetic and chemical behaviors that deviate from those of the bulk [4,32,36,73,76,90,92,97,98,100,105,111,112,113,114,115,116].

#### 4.6.1. Surface Spin Disorder

One key phenomenon is surface spin disorder, which arises from the asymmetrical coordination of surface atoms. This lack of balanced magnetic interactions leads to spin canting or disorder at the surface, reducing the net magnetization compared to the MNP’s bulk counterpart. Surface spins contribute less to the overall magnetic order, diminishing the NP’s saturation magnetization [4,32,76,97,98,115].

#### 4.6.2. Enhanced Surface Magnetic Anisotropy

Surface atoms exist in different chemical and magnetic environments compared to core atoms, resulting in higher magnetic anisotropy at the surface. This heightened anisotropy impacts the stability of magnetic moments, often requiring more energy to reorient these moments, which can influence the particle’s magnetic behavior under applied fields [4,32,76,97,105,115].

#### 4.6.3. Surface Reactivity and Degradation

The MNPs’ high surface area also makes them more chemically reactive. This reactivity can lead to unwanted oxidation or degradation, especially in reactive metallic MNPs such as iron or cobalt. These effects alter the magnetic properties of the nanoparticles and can compromise their stability and performance in practical applications [4,32,76,92,97,115].

#### 4.6.4. Magnetic Dead Layer

Moreover, in some cases, the surface atoms may not contribute to the MNP’s magnetization, forming a magnetic dead layer. This dead layer effectively reduces the overall MNP saturation magnetization ${\left[M_{sat}\right]}_{MNP}$, as described in Equation (45) [4,32,76,97,98,112,113,114,115,116,117]:(45)$${\left[M_{sat}\right]}_{MNP}={\left[M_{sat}\right]}_{Bulk}{\left(1-\frac{2w_{DL}}{d}\right)}^{3},$$
where $w_{DL}$ is the deal layer width, d is the diameter of the MNP, and ${\left[M_{sat}\right]}_{Bulk}$ is the saturation magnetization of the MNP bulk material. This relationship highlights that for smaller particles, even a relatively thin dead layer can significantly reduce the overall magnetization due to the increased proportion of the non-magnetic volume.

#### 4.6.5. Exchange Bias

For MNPs with an FM core covered with an AFM shell (or vice versa), the magnetic interactions at the FM/AFM interface result in a phenomenon known as exchange bias. This interaction manifests as a shift in the M(H) hysteresis curve along the field axis. The magnitude of the exchange bias depends on factors such as the interface quality, particle size, and the strength of the FM/AFM coupling. This shift can be utilized in applications where controlled magnetic behavior is essential, such as spintronic devices and magnetic sensors [4,32,73,76,97,111,115].

## 5. Routine Characterization Techniques for MNPs

The study of MNPs can involve numerous characterization techniques, ranging from routine methods [16,32,86,87,108,118,119,120,121,122,123,124,125,126] to highly sophisticated ones [16,120,121,122,124,127,128,129,130], depending on the research objectives and available facilities. However, certain widely accessible techniques are commonly used in most MNP studies. These include X-ray Diffraction (XRD), Raman Spectroscopy, Scanning and Transmission Electron Microscopy (TEM and STEM), often coupled with Energy-dispersive X-ray Spectroscopy (EDS) and magnetometry, with the latter being particularly emphasized [16,87,118,119,122,123,124,126].

### 5.1. XRD

XRD is a widely utilized, non-destructive, and generally accessible technique to characterize MNPs. In the context of MNPs, XRD is instrumental in confirming the formation of the desired crystallographic phases, assessing the level of crystallinity, and identifying any secondary phases that may be present.

Furthermore, for single-domain MNPs, the Scherrer equation can be applied to estimate the average particle size, $\langle d_{XRD}\rangle$, as expressed in Equation (46) [32,92,122,124,131]:(46)$$\langle d_{XRD}\rangle =\frac{k_{XRD}\cdot \lambda_{XRD}}{\beta_{XRD}\cdot cos\theta },$$
where $\beta_{XRD}$ represents the full width at half maximum (FWHM) in radians of a selected diffraction peak (typically the main peak), measured in a Bragg–Brentano configuration; θ is the diffraction angle of the considered plane, $\lambda_{XRD}$ is the X-ray wavelength, and $k_{XRD}$ is a shape factor, typically approximated as 0.9 for spherical crystallites. This equation provides an estimate of the average crystallite size, which, for single-domain MNPs, often matches well to sizes determined by TEM. Thus, XRD offers a quick upper-limit estimate of particle size.

However, as indicated by Equation (46), very small MNPs exhibit broad diffraction peaks, which can make phase identification ambiguous, especially when the MNPs belong to the same crystallographic group. This limitation often complicates distinguishing between different iron oxides or ferrites based solely on XRD diffractograms. To address this, complementary characterization techniques should always be employed.

Furthermore, most XRD setups use Cu $K_{\alpha }$ radiation, which often causes elevated background levels from Fe fluorescence when analyzing Fe-containing samples. This can complicate phase identification, particularly for materials such as magnetite, maghemite, or their mixtures. While fluorescence can be suppressed using monochromators, this approach reduces peak intensity and penetration depth, potentially leading to inconclusive diffractograms. Alternatively, adopting a Co radiation source provides a more effective solution for iron-based MNPs, as it significantly reduces fluorescence, resulting in an improved peak-to-noise ratio and more reliable phase identification [132].

### 5.2. Raman Spectroscopy

Raman spectroscopy, like XRD, is a widely used and accessible nondestructive technique to characterize MNPs. It is particularly valued for its ability to identify material-specific fingerprints, making it an effective tool to determine the phases present in a sample [16,31,118,126,133,134].

Raman spectroscopy is especially useful in the analysis of ferrite MNPs, one of the most commonly studied types of MNPs. Unlike XRD, Raman spectroscopy is less affected by size-induced broadening of spectral features, allowing for a more precise differentiation between various ferrite types [118,126,134,135]. Furthermore, Raman spectra can provide insights into the inversion parameter of ferrites, a structural characteristic that is often challenging to determine using XRD due to the substantial peak broadening observed in smaller particles, as noted in Equation (46) [118,126].

This complementary role makes Raman spectroscopy a valuable tool in the comprehensive characterization of MNPs.

### 5.3. Transmission Electron Microscopy and EDS

Transmission electron microscopy, including STEM and high-resolution TEM (HRTEM), is a particularly powerful tool to characterize MNPs. These techniques are particularly valuable when combined with EDS, as they enable almost direct visualization of individual NPs. This capability allows for the precise determination of size, shape, and spatial distribution of MNPs, while also providing both qualitative and quantitative elemental distribution at atomic resolution.

Atomic-scale imaging and elemental mapping are especially advantageous to characterize MNPs with complex core–shell or doped structures. These methods can reveal variations in composition within a particle, such as differences between the core and shell regions, and can also help estimate the width of magnetic dead layers.

HRTEM, in particular, offers highly detailed images of the crystal structure, including lattice fringes. Such information is critical to assess the crystallinity of individual MNPs and analyze their atomic arrangements, which are essential to understand and optimize their magnetic properties.

To determine the size distribution of MNPs from TEM/STEM images, one should tally the diameters of a statistically significant sample of nanoparticles, create a histogram by grouping particles into size bins, and then adjust a log-normal function to fit the MNPs’ size distribution, f(d), using Equation (47) [46,92,115,119]:(47)$$f\left(d\right)=\frac{1}{d\cdot \sigma_{nd}\sqrt{2\pi }}e^{\left(-\frac{{\left(\ln d-\mu_{nd}\;\right)}^{2}}{2\sigma_{nd}^{2}}\right)},$$
where d is the MNPs diameter, and $\mu_{nd}$ and $\sigma_{nd}$ are the mean and standard deviation of the diameter’s natural logarithm. Therefore, it is possible to determine the average MNP diameter estimated using TEM/STEM using Equation (48):(48)$$\langle d_{TEM}\rangle =E\left[d\right]=e^{\left(\mu_{nd}+\frac{\sigma_{nd}^{2}}{2}\right)},$$

and its standard deviation using Equation (49):(49)$$\sigma_{d_{TEM}}=\sqrt{Var\left[d\right]}=\sqrt{\left[e^{\left(\sigma_{nd}^{2}-1\right)}\right]e^{\left(2\mu_{nd}+\sigma_{nd}^{2}\right)}}.$$

Finally, the homogeneity of a given sample can be assessed by calculating the polydispersity index (PDI), which is defined by Equation (50) [1,46,83,136,137,138]:(50)$$PDI={\left(\frac{\sigma_{d_{TEM}}}{\langle d_{TEM}\rangle }\right)}^{2}.$$

A PDI close to zero indicates a monodisperse sample, with particles of nearly uniform size, while higher values reflect greater variability in particle dimensions, signifying increased polydispersity. This metric is especially critical in applications or synthesis processes where particle uniformity is essential.

### 5.4. Magnetometry

Vibrating Sample Magnetometry (VSM) and Superconducting Quantum Interference Device (SQUID) magnetometry are essential tools to characterize the magnetic properties of MNPs. By measuring the magnetization of MNPs under varying magnetic fields and temperatures, these techniques provide comprehensive properties, including $M_{sat}$, $M_{r}$, $H_{c}$, $T_{C}$, and $T_{B}$. These properties are critical to tailor MNPs to meet the specific requirements of diverse applications ranging from biomedical to industrial, making magnetometry indispensable in MNPs research [16,122,123,124].

Magnetic properties, such as χ, $M_{sat}$, $M_{r}$, $H_{c}$, and $T_{C}$, can be readily extracted or fitted from M(H) and M(T) curves, following the definitions provided in the previous sections.

Temperature-dependent magnetization studies, such as Zero-Field-Cooled (ZFC) and Field-Cooled (FC) curves (illustrated in Figure 10), provide essential insights into SPM properties, particularly to estimate $T_{B}$ and $d_{B}$.

A ZFC curve is obtained by cooling the sample to the lowest measured temperature in the absence of an applied magnetic field. Once at the minimum temperature, a small magnetic field (typically 1–10 mT) is applied, and the magnetization measured as the temperature is gradually increased to the maximum desired value.

An FC curve follows a similar protocol but applies a magnetic field during the cooling stage. Usually, the same magnetic field is maintained during the subsequent heating stage when the magnetization is recorded.

In real systems, typically there is a distribution of particle sizes and anisotropy energy barriers. To estimate $T_{B}$ based on ZFC-FC curves, several approaches can be applied.

The simplest and most used method is to identify the temperature at which the ZFC curve reaches its maximum value, which is often referred to as the MNPs’ average blocking temperature, $\langle T_{B}\rangle$. This temperature corresponds to the point where thermal energy is sufficient to overcome the average anisotropy energy barrier of the particles, enabling them to align with the applied field. At this point, the net magnetization reaches its peak because the majority of particles transition from the blocked state to the SPM state.

By applying Equation (44), we can use $\langle T_{B}\rangle$ to estimate the expected average MNPs diameter:(51)$$\langle d_{T_{B}}\rangle =\sqrt[{3}]{\frac{6k_{B}\langle T_{B}\rangle \cdot ln\left(\frac{\tau_{m}}{\tau_{0}}\right)}{\pi \cdot K_{\mathrm{eff}}}}\approx \sqrt[{3}]{\frac{150k_{B}\langle T_{B}\rangle }{\pi \cdot K_{\mathrm{eff}}}}.$$

Another relevant temperature is the temperature of irreversibility, $T_{ir}$, which is characterized by the point where the FC and ZFC curves diverge when observed from higher to lower temperatures. $T_{ir}$ marks the onset of non-reversible behavior between the two measurements, indicating that below this temperature, the thermal energy is no longer sufficient to keep all the particles in the SPM state. As a result, larger MNPs begin to transition into the blocked state. Therefore, the irreversibility temperature can be used to estimate the highest possible $T_{B}$, $T_{ir}={\left\{T_{B}\right\}}_{max}$.

Similarly to $\langle d_{T_{B}}\rangle$, it is also possible to estimate the diameter of the largest MNPs, ${\left\{d_{T_{B}}\right\}}_{max}$, using Equation (52):(52)$${\left\{d_{T_{B}}\right\}}_{max}=\sqrt[{3}]{\frac{6k_{B}{\left\{T_{B}\right\}}_{max}\cdot ln\left(\frac{\tau_{m}}{\tau_{0}}\right)}{\pi \cdot K_{\mathrm{eff}}}}\approx \sqrt[{3}]{\frac{150k_{B}{\left\{T_{B}\right\}}_{max}}{\pi \cdot K_{\mathrm{eff}}}}.$$

More sophisticated approaches can be employed to estimate the distribution of $T_{B}$, $f_{T_{B}}(T)$. Wohlfarth suggested that this could be estimated by differentiating the product of the temperature and the ZFC magnetization, $M_{ZFC}$, as described by Equation (53) [139,140]:(53)$$f_{T_{B}}\left(T\right)\propto \frac{d\left({T\cdot M}_{ZFC}\right)}{dT}.$$

Still, this approach neglects the contribution of blocked particles.

A more popular method, presented in several works, is described by Equation (54) [104,115,140,141]:(54)$$f_{T_{B}}\left(T\right)\propto \frac{d\left(M_{ZFC}-M_{FC}\right)}{dT}.$$

However, some studies, such as the ones reported by Tournus et al. [140], argue that the approach from Equation (54) lacks an additional −1T multiplicative term. To address this, they propose the following modified description [140,141,142]:(55)$$f_{T_{B}}\left(T\right)\propto -\frac{1}{T}\frac{d\left(M_{ZFC}-M_{FC}\right)}{dT}.$$

It is important to note that $T_{B}$ is solely sensitive to the magnetic core of the MNPs. Therefore, if the MNPs are coated or functionalized with a diamagnetic or paramagnetic layer, this outer shell will not contribute to $T_{B}$. This also applies to the dead layer, hence if it is too thick. The average particle size determined by TEM ($\langle d_{TEM}\rangle$) may substantially differ from $\langle d_{T_{B}}\rangle$. Still, for MNPs with a negligible dead layer thickness, $\langle d_{TEM}\rangle \approx \langle d_{T_{B}}\rangle$. In this case, knowing $\langle T_{B}\rangle$ allows us to rearrange Equation (51) to estimate $K_{\mathrm{eff}}$, which is especially useful for new materials where the order of magnitude of $K_{\mathrm{eff}}$ may not be known.

Another way to estimate $K_{\mathrm{eff}}$ using magnetometry is by measuring the dependence of $H_{c}$ as a function of temperature, as described by Equation (56) [143]:(56)HcB=α2KeffMsat1−TTB12,
where ${\left(H_{c}\right)}_{B}$ is the coercive field of the blocked MNPs (for SPM MNPs $H_{c}=0$), and α=1 if the MNPs’ easy axis are aligned with H, or α=0.48 if they are randomly oriented. For SPM particles homogeneously dispersed in a medium or in powder form, it is reasonable to consider that α=0.48. Thus, by rearranging Equation (56), it is possible to obtain $K_{\mathrm{eff}}$.

#### Additional Considerations Regarding Magnetometry

Magnetometry is an exceptionally versatile technique to characterize MNPs, as demonstrated in the previous section. However, certain precautions are essential, particularly regarding the accuracy of the absolute M measured.

The geometry of the sample and any radial offset from the optimal measuring position can significantly impact the measured magnetization values, as highlighted in several studies [144,145,146,147]. It is important to note that these effects are not related to the demagnetizing field (they exist even for null demagnetizing fields). To minimize these issues, it is advisable to use sample geometries and mounting conditions consistent with a reference sample of well-known magnetization. For magnetometers employing a second-order gradiometer, such as the MPMS3 from Quantum Design, a straightforward universal empirical correction method is available, as reported by Amorim et al. [144].

Additionally, the magnet used to apply the magnetic field may retain remanent fields after applying high fields (typically greater than 1 T). This residual field can affect the accuracy of $H_{c}$ estimation. To address this, it is crucial to determine the actual applied field (as the displayed field often derives from a calibration curve) and adjust the measurement parameters accordingly.

Finally, to determine $T_{B}$, the applied magnetic field during the ZFC/FC measurements must be weak enough to avoid substantially altering the energy barriers of the particles.

## 6. Magnetic Induced Heating

In addition to Néel relaxations, single-domain MNPs that can mechanically rotate within a given medium (e.g., a liquid). These are referred to as Brown relaxations, as illustrated in Figure 8, and their characteristic relaxation time is given by Equation (57) [1,45,46,47,76,92,105]:(57)$$\tau_{B}=\frac{3\eta V_{H}}{k_{B}T},$$
where η is the viscosity of the surrounding medium, and $V_{H}$ is the hydrodynamic volume of the MNP.

Assuming that Néel and Brown relaxation mechanisms are independent, the effective relaxation time (τ) can be described by Equation (58) [1,45,46,47,76,92,105]:(58)$$\frac{1}{\tau }=\frac{1}{\tau_{N}}+\frac{1}{\tau_{B}}⟺\tau =\frac{\tau_{N}\tau_{B}}{\tau_{N}+\tau_{B}}.$$

From this relationship, it is evident that when $\tau_{N}$ is significantly larger than $\tau_{B}$, the effective relaxation time is dominated by Brown relaxations. Conversely, when $\tau_{B}$ is much larger than $\tau_{N}$, the Néel relaxations dominate. This behavior is illustrated in Figure 11a, with the inset highlighting the transition between these regimes.

Rosensweig demonstrated that SPM NPs exposed to alternating magnetic fields (AMFs) dissipate power due to both Néel and Brown relaxation mechanisms, as described by Equation (59) [1,45,46,47,76,92,105]:(59)$$P=\frac{1}{2}\mu_{0}\cdot \chi_{0}\cdot \omega \cdot H_{0}^{2}\cdot \frac{\omega \cdot \tau }{1+{\left(\omega \cdot \tau \right)}^{2}},$$
where $\chi_{0}$ is the static magnetic susceptibility, τ is the effective relaxation time, and $H_{0}$ and $\omega =2\pi f_{AMF}$ are the amplitude and angular frequency of the applied AMF, respectively.

From this equation, it is evident that the term inside the fraction determines the shape of the dissipated power as a function of the AMF frequency and the nanoparticle size (which strongly influences τ), as illustrated in Figure 11b.

The term outside the fraction acts as a scaling factor. This factor exhibits a linear dependence on $\chi_{0}$ and ω, but a quadratic dependence on $H_{0}$, which is also illustrated in Figure 11c,d. These dependencies highlight the critical role of both the AMF parameters and MNP characteristics in the resulting power dissipation.

### 6.1. Magnetic Hyperthermia

Magnetic hyperthermia is a promising experimental cancer treatment that leverages the heat generated by MNPs under alternating AMF to elevate the temperature of cancerous tissues to 42–47 °C. This temperature range is sufficient to selectively damage or destroy cancer cells, which are more sensitive to heat than healthy cells [5,8,17,19,20,21,22,23,25,28,42,47,49,50,148,149,150].

In systemic or intra-tumoral applications, it is preferable to minimize the amount of MNPs introduced into the body, even if they exhibit low toxicity. Therefore, maximizing the heat generated per unit mass of MNPs is critical to ensure the effectiveness of the treatment. This efficiency is quantified by the specific absorption rate (SAR), also referred to as the specific loss power (SLP), which is defined by Equation (60) [1,45,46,47,76,92,105]:(60)$$SLP=SAR=\frac{P}{m_{MNP}},$$
where P is the power dissipated by the MNPs, as described in Equation (59), and $m_{MNP}$ is their mass.

Several strategies can be employed to maximize the SLP of a given amount of MNPs. To begin, MNPs should be synthesized using a magnetic material with the highest possible $\chi_{0}$. Moreover, the AMF frequency and MNP size should be optimized to maximize the fractional term in Equation (59). Finally, the $H_{0}$ and $f_{AMF}$ of AMF should be set as high as possible.

However, in clinical settings, safety constraints must be considered to avoid overheating normal tissues due to eddy currents. To comply with these requirements, several guidelines were proposed, namely the widely recognized Atkinson–Brezovich limit that states that [151](61)$${\left[f_{AMF}\cdot H_{0}\right]}_{Atkinson–Brezovich\;}\leq 4.85\times 10^{8}\;A\;m^{-1}\;s^{-1}.$$

Notwithstanding, several studies suggest that higher limits may also be feasible [35,96,152]. Garcia-Alonso et al. propose that [153](62)$${\left[f_{AMF}\cdot H_{0}\right]}_{Garcia-Alonso\;}\leq 9.59\times 10^{9}\;A\;m^{-1}\;s^{-1},$$
while Kim et al. note that current MRI systems, operating at 500 MHz and 11.7 T, already reach [154](63)$${\left[f_{AMF}\cdot H_{0}\right]}_{\mathrm{MRI}@500\;MHz\;and\;11.7\;T\;}=4\times 10^{10}\;A\;m^{-1}\;s^{-1},$$
therefore, they suggest that, for short pulses, this limit could increase to [155](64)$${\left[f_{AMF}\cdot H_{0}\right]}_{Kim\;}\leq 7\times 10^{11}\;A\;m^{-1}\;s^{-1}.$$

To maximize the SLP while adhering to these constraints, a practical approach is to prioritize increasing $H_{0}$ over $f_{AMF}$, since $H_{0}$ contributes quadratically to the SLP, while $f_{AMF}$ only has a linear contribution, as illustrated in Figure 11c,d.

Determining the SLP using the P described in Equation (59) is an effective theoretical approach to plan and/or simulate expected results. However, its accurate experimental determination is often challenging. Consequently, a calorimetric approach is typically adopted. In this method, MNPs are dispersed in a medium resembling their intended application, and the temperature is monitored over time under a given AMF. The SLP is then estimated using Equation (65) [45,76,105,149]:(65)$$\begin{array}{l} SLP=SAR=c_{total}{\left(\frac{dT}{dt}\right)}_{0}=\\ =\left[\frac{m_{MNP}}{m_{MNP}+m_{med}}c_{MNP}+\frac{m_{med}}{m_{MNP}+m_{med}}c_{med}\right]{\left(\frac{dT}{dt}\right)}_{0}\begin{array}{c} m_{MNP}\ll m_{total}\\ \approx \end{array}c_{med}{\left(\frac{dT}{dt}\right)}_{0}, \end{array}$$
where ${\left(\frac{dT}{dt}\right)}_{0}$ is the initial slope of the temperature versus time curve, $m_{total}=m_{MNP}+m_{med}$ is the total mass, $m_{MNP},$ and $m_{med}$ are the MNPs and medium masses, respectively, and $c_{total},\;c_{MNP}$, and $c_{med}$ are their specific heat capacities.

As noted, both $H_{0}$ and $f_{AMF}$ scale the SLP, which means that comparing SLP values across studies can be misleading. For the same material, different SLP values can be measured depending on the AMF applied. To avoid artificially inflated SLP values, the Intrinsic Loss Power (ILP) is used [45]:(66)$$ILP=\frac{SLP}{{f_{AMF}\cdot H}_{0}^{2}}.$$

It is a standardized parameter that enables a fair comparison of the heating efficiency of different MNPs, regardless of the AMF amplitude and frequency applied. However, it is important to note that maximizing the fractional term of Equation (59) (by tuning ω·τ) is essential to optimize the ILP. This underlines the necessity of selecting an appropriate $f_{AMF}$ to optimize the effective relaxation dynamics of MNPs with specific sizes and magnetic anisotropies.

### 6.2. Self-Regulated Heating

Magnetic hyperthermia offers clear advantages over conventional hyperthermia techniques, notably its low invasiveness and the absence of ionizing radiation from AMFs. Another particularly promising feature is its potential for self-regulated heating. Because MNPs are typically FM or FiM materials, they exhibit a $T_{C}$ above which their $\chi_{0}$ drastically decreases to nearly zero. This characteristic allows for the design of MNPs with a tailored $T_{C}$ and a M(T) profile such that, upon reaching a critical temperature near $T_{C}$, the SLP diminishes significantly. As a result, the system inherently prevents overheating, maintaining the temperature within the targeted hyperthermia range, regardless of local MNPs concentration or the applied field strength and frequency of the AMF [105,107,110,119,148,149].

This self-regulating behavior effectively acts as a safety mechanism, enhancing patient safety and making magnetic hyperthermia highly suitable for medical applications. Furthermore, the $T_{C}$ of ferrite-based MNPs can often be conveniently tuned through doping or substitution of transition metals, providing additional flexibility in optimizing their thermal response for specific therapeutic needs [21,102,104,110,111,112,113].

## 7. Concluding Remarks

This review has provided a comprehensive overview of the key concepts, magnetic properties, and characterization techniques essential to understand and advance the study of MNPs. Aimed at both newcomers and experienced researchers, it bridges the foundational principles of nanomagnetism with the practical insights needed to design, analyze, and utilize MNPs effectively.

Central to this discussion is the unique magnetic behavior of MNPs, including superparamagnetism, single-domain behavior, and the influence of surface effects such as spin disorder and the magnetic dead layer. These phenomena arise due to the interplay of nanoscale effects, magnetic anisotropy, and thermal energy, which collectively determine critical parameters such as $T_{B}$ and coercivity $H_{c}$. The review also explored how these properties differentiate MNPs from their bulk counterparts, making them versatile for a wide range of applications.

Characterization techniques such as XRD, Raman spectroscopy, and electron microscopy (TEM, STEM, and EDS) were detailed, showcasing their utility in determining structural and compositional properties. Still, the focus was placed on magnetometry as the cornerstone technique to characterize the magnetic properties of MNPs. The utility of M(H) and M(T) curves, as well as ZFC and FC studies, was highlighted in determining essential parameters such as $M_{sat}$, $M_{r},$ and $T_{B}$. Furthermore, guidance was provided to avoid common pitfalls in VSM and SQUID magnetometry measurements, ensuring that readers are equipped with the necessary precautions to obtain reliable and accurate results.

While synthesis methods and material selection were only briefly addressed, the focus remained on their impact on magnetic properties rather than the specifics of synthetic protocols. Nonetheless, relevant references are provided to enable readers to delve deeper into these less-explored topics. Similarly, applications such as magnetic hyperthermia were discussed in the context of their reliance on specific magnetic phenomena, including relaxation mechanisms and self-regulated heating. Safety and efficiency considerations for clinical use, such as the Atkinson–Brezovich limit and its alternatives, were also explored.

In summary, this review has synthesized the most relevant topics in MNP research, focusing on their magnetic behaviors, characterization methodologies, and functional applications. Ultimately, it seeks to bridge the gap between introductory material and advanced research, providing a balanced perspective that caters to a broad audience. Through careful curation and clear exposition, I hope to inspire and empower a new generation of scientists to explore the exciting possibilities that MNPs have to offer.

## Figures and Tables

**Figure 1 pharmaceutics-17-00137-f001:**
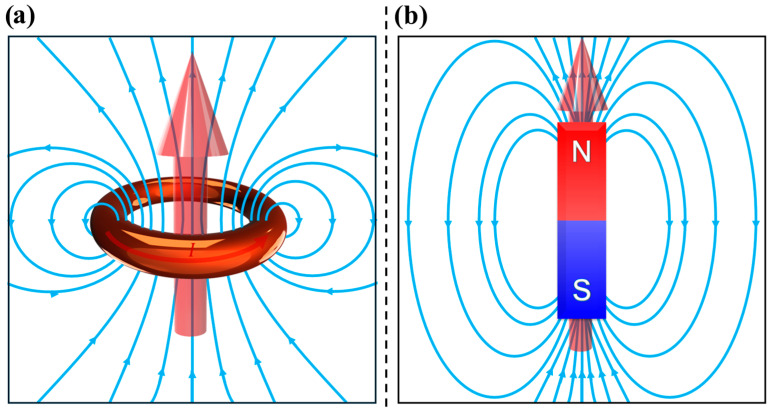
Representation of a magnetic dipole moment (red arrow) and corresponding magnetic field lines when generated by (**a**) an electric current circulating within a loop of area A and (**b**) a bar magnet with magnetic (mono)poles N and S.

**Figure 2 pharmaceutics-17-00137-f002:**
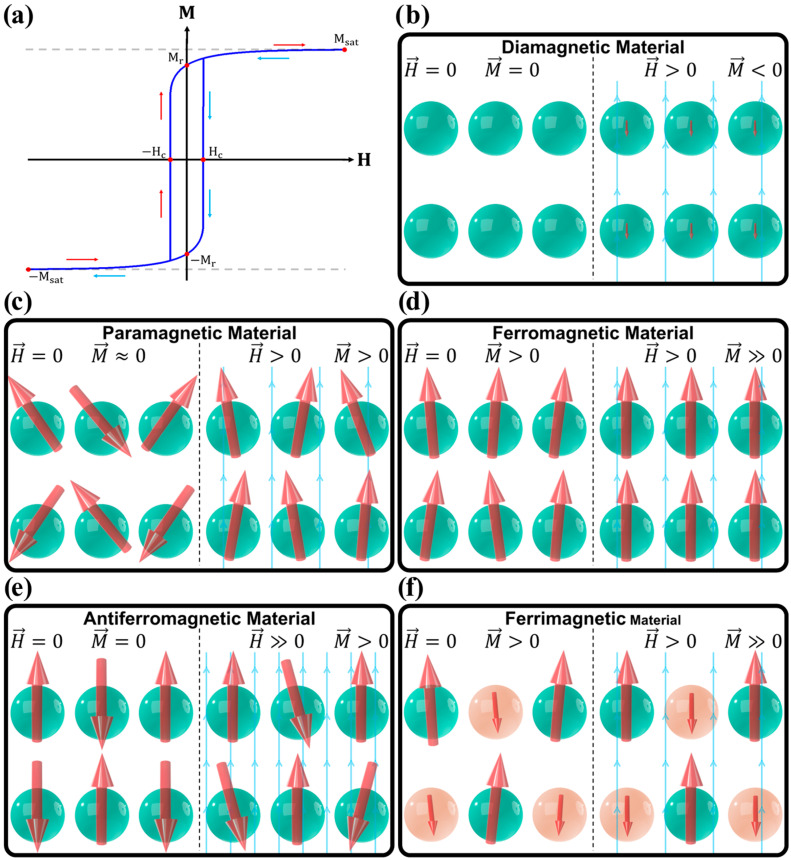
(**a**) Key magnetic properties derived from an M(H) hysteresis curve. (**b**–**f**) Magnetic material classifications showing magnetic moment arrangements with and without an applied field H. Each ball represents an atom/ion and different color atoms represent different types of atoms/ions.

**Figure 3 pharmaceutics-17-00137-f003:**
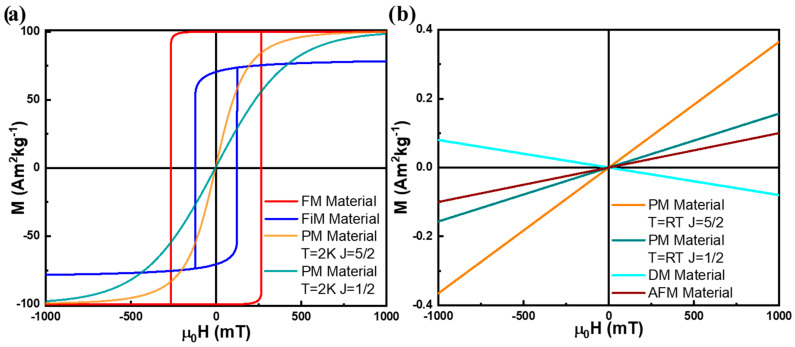
Illustrative M(H) hysteresis curves for (**a**) a FM material, a FiM material, and two different PM materials with J=12 and J=52 at 2 K; (**b**) a DM material, an AFM material, and two different PM materials with J=12 and J=52 at 300 K (around room temperature).

**Figure 4 pharmaceutics-17-00137-f004:**
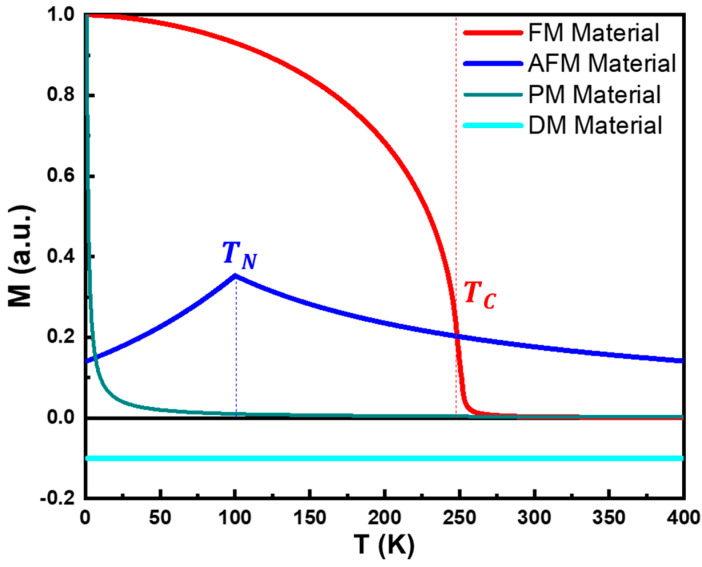
Illustrative M(T) curves of a DM material, a PM material, an FM material with $T_{C}\approx 250\;K$, and an AFM with $T_{N}\approx 100\;K$.

**Figure 5 pharmaceutics-17-00137-f005:**
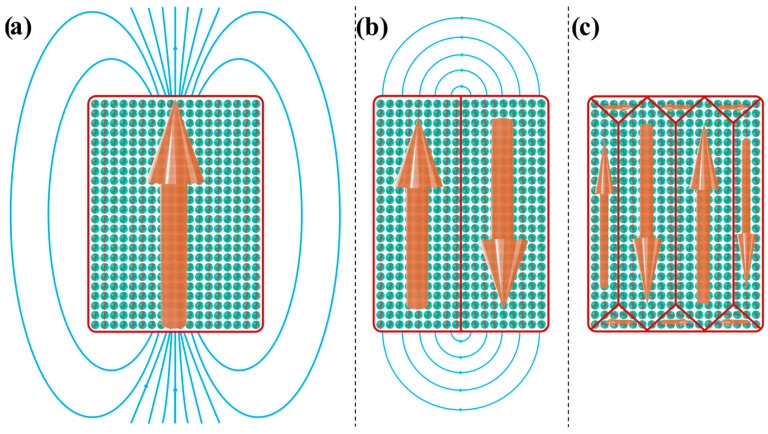
Formation of magnetic domains in an FM material to minimize the magnetic energy associated with magnetic stray fields. (**a**) With a single magnetic domain, the abundance of stray fields maximizes the magnetostatic energy. (**b**) The creation of two opposing magnetic domains visibly reduces stray fields. (**c**) Further domain formation reduces the stray fields close to zero.

**Figure 6 pharmaceutics-17-00137-f006:**
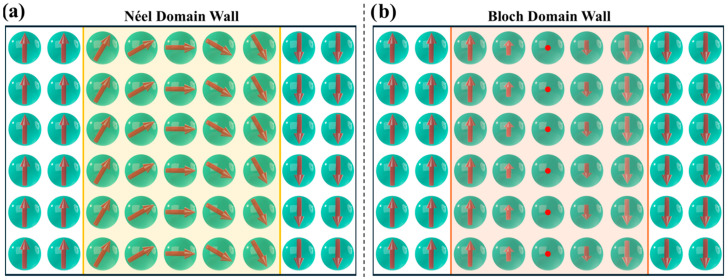
Types of magnetic domain walls formed in FM and FiM materials: (**a**) Néel Domain Wall and (**b**) Bloch Domain Wall.

**Figure 7 pharmaceutics-17-00137-f007:**
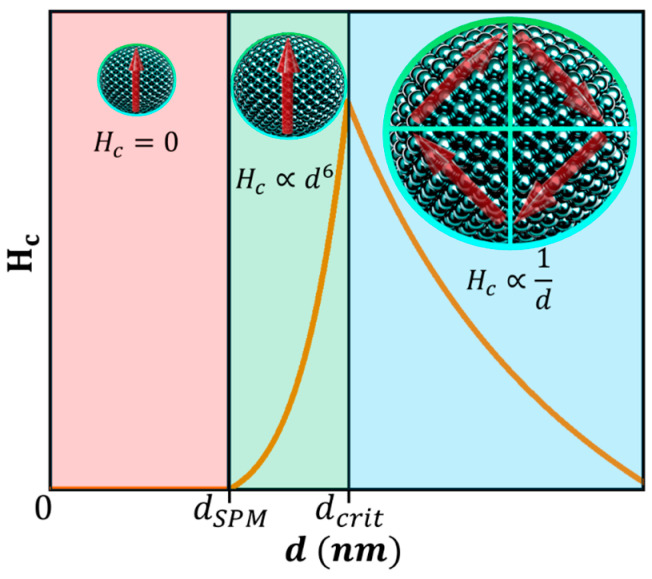
MNPs size-dependent coercivity for three different regimes: single-domain SPM NP, single-domain-locked FM NP, and multiple-domain FM NP.

**Figure 8 pharmaceutics-17-00137-f008:**
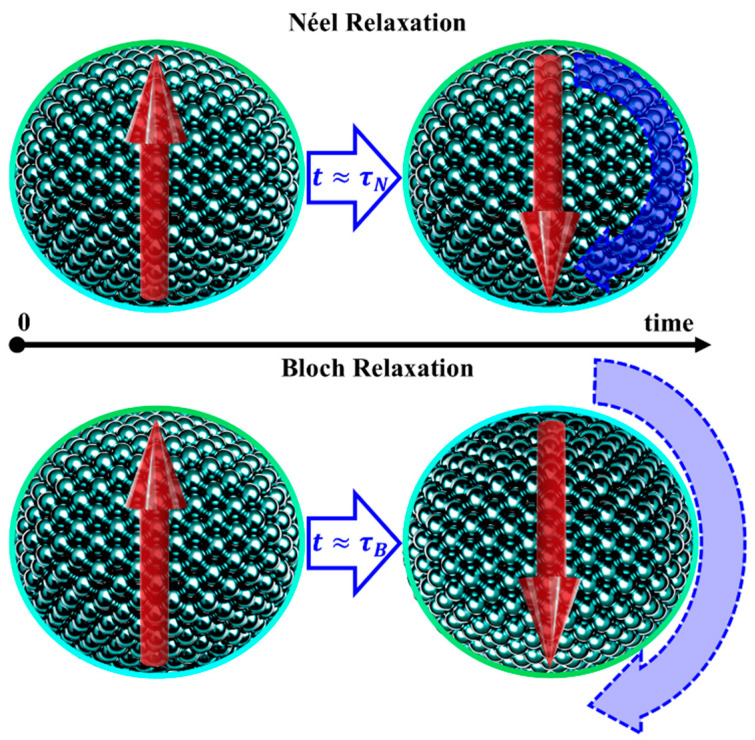
Néel and Brown relaxation mechanisms. While in the Néel relaxation, the magnetic moment rotates without particle rotation. In the Brown relaxation, the particle physically/mechanically rotates along with the magnetic moment.

**Figure 9 pharmaceutics-17-00137-f009:**
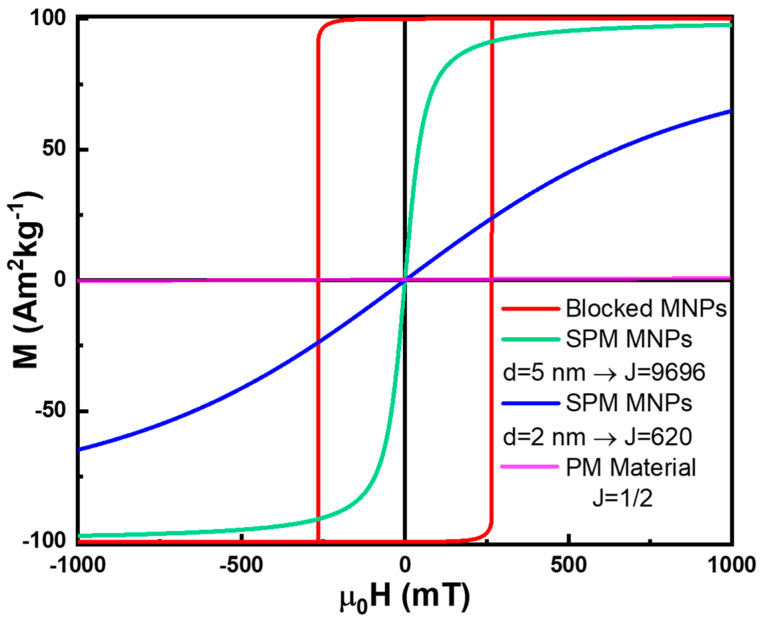
M(H) curves of MNPs in the PM, SPM, and blocked states. For the SPM NPs, two different diameters were considered, resulting in different $J_{MNP}=N\cdot J_{atom}$, where $J_{atom}$ = 2, and N was calculated based on the particle volume and its atomic density (assumed to match metallic Fe).

**Figure 10 pharmaceutics-17-00137-f010:**
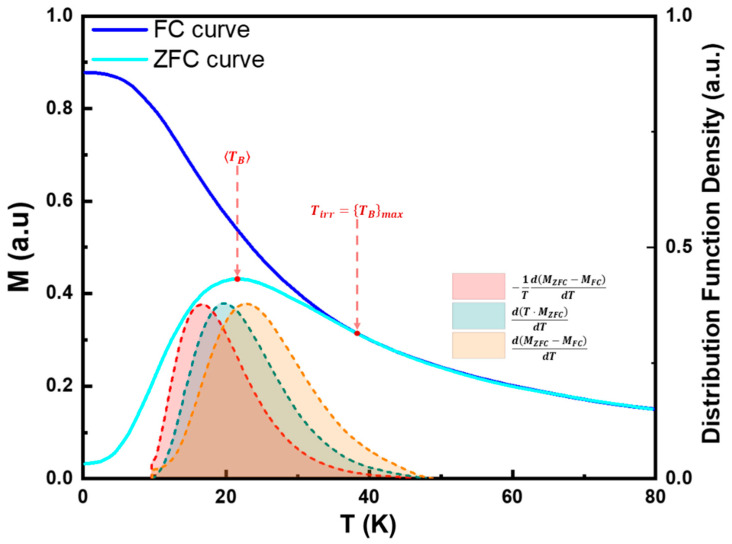
Illustration of FC and ZFC M(T)M(T) curves, highlighting the properties derived from their analysis. Colored areas on the curves represent various methods to estimate $T_{B}$ distributions.

**Figure 11 pharmaceutics-17-00137-f011:**
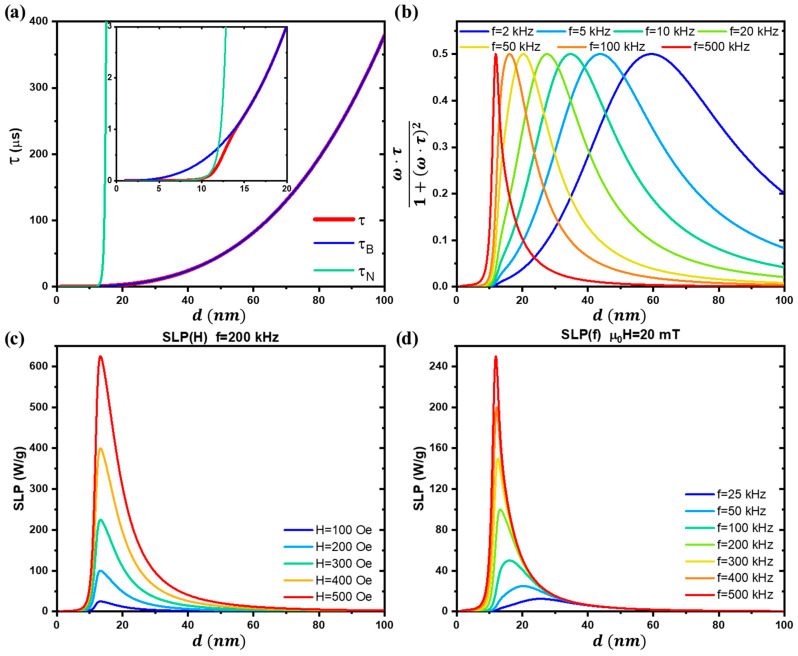
(**a**) Illustration of the Néel, Brown and effective relaxation times. The inset provides a closer look at the crossover region, where the effective relaxation time transitions from Néel-dominated to Brown-dominated relaxation. (**b**) Variation in SLP curve shape, determined by the fractional term of Equation (59), as a function of MNP diameter for different AMF frequencies. (**c**) Effect of AMF amplitude on SLP, showing a quadratic dependence. (**d**) Effect of AMF frequency on SLP, demonstrating a linear dependence.

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
