# Peer review of "A Compendium of Magnetic Nanoparticle Essentials: A Comprehensive Guide for Beginners and Experts"

_pharmaceutics, 2025, doi:10.3390/pharmaceutics17010137_

Round 1

Reviewer 1 Report (Previous Reviewer 1)

Comments and Suggestions for Authors Dear author, this version of the article contains a lot of similar editorial errors.
"Error! Reference source not found." This phrase is spred all over the text.
Please, fix it!

Reviewer 2 Report (Previous Reviewer 2)

Comments and Suggestions for Authors

The manuscript has improved substantially with the revision. No doubt the citations have been inserted with a publisher and, in particular books, may not have a DOI although it may not have been included in the database. In any case, after working on the proofs, readers should not have difficulty locating the texts.

This manuscript is a resubmission of an earlier submission. The following is a list of the peer review reports and author responses from that submission.

Round 1

Reviewer 1 Report

Comments and Suggestions for Authors

The outstanding article "A Compendium of Magnetic Nanoparticle Essentials: A Comprehensive Guide for Beginners and Experts" has almost nothing in common with Pharmaceutics journal.

It seems that the Author wrote the main part of the paper and then slightly edited the introduction to fit the high-ranking Pharmaceutics journal.

One of the Pharmaceutics statements is: "The full experimental details must be provided so that the results can be reproduced. The reviewed article is absolutely theoretical. I'd would recomend to publish it in some physics journal.

The main part thoroughly describes theory and methods used to study magnetic nanoparticles. The abstract says that emphasis was placed on routine characterization methods, including X-ray diffraction...

However, not a single diffraction pattern was provided in the article. Analyzing an iron containing sample with copper radiation (which is commonly used for powder XRD) yields a high background level due to fluorescence. This can lead to incorrect phase identification. It is recommended to use cobalt radiation to reduce iron fluorescence [Yvonne M. Mos, Arnold C. Vermeulen, Cees J. N. Buisman & Jan Weijma (2018) X-Ray Diffraction of Iron Containing Samples: The Importance of a Suitable Configuration, Geomicrobiology Journal, 35:6, 511-517, DOI: 10.1080/01490451.2017.1401183]. I insist on covering this issue in an article, whether it is published or not.

Reviewer 2 Report

Comments and Suggestions for Authors

The manuscript pharmaceutics-3355463 represents a very a thorough review of the current status of the magnetic properties in nanoparticles, from all the multifacets of interest. I consider that this review is an improvised work, which has required a notable effort for the author.

It is also a difficult work to appreciate for many potential readers (including reviewers).

I honestly think if Pharmaceutics is the most appropriate journal, among those covered by MDPI, although I do not consider it correct from the point of view of the applicability of relevant facets...

Reading the text, I find an excessive use of phrases written in bold characters, as well as the identification of the equations (in parentheses).

I suggest the author refer to Point 7 as 'Concluding remarks'.

Misprints:

Line 103: (equation (3)), should be said (equation (3),

Line 140: equations (9) and (9) should be said equations (9) and (10)
